

# Mixed Phase Orographic Cloud Microphysics during StormVEx and IFRACS

Douglas H. Lowenthal[1], A. Gannet Hallar[1,2], Robert O. David[3],
Ian B. McCubbin[1], Randolph D. Borys[1], and Gerald G. Mace[2]

[1]Desert Research Institute, 2215 Raggio Pkwy., Reno, NV 89509
[2]University of Utah, 135 S 1460 E, Salt Lake City, UT 84112
[3]ETH Zürich, Universitätstrasse 16, 8092 Zürich, Switzerland

*Correspondence to*: Douglas Lowenthal (dougl@dri.edu)

**Abstract.** Wintertime mixed phase orographic cloud (MPC) measurements were conducted at the Storm Peak Laboratory (SPL) during the StormVEx and IFRACS programs in 2011 and 2014, respectively. The data include 92 hours of simultaneous measurements of supercooled liquid cloud droplet and ice particle size distributions (PSD). Average cloud droplet number concentration (CDNC), droplet size (NMD) and liquid water content (LWC) were similar in both years while ice particle concentration (Ni) and ice water content (IWC) were higher during IFRACS. The consistency of the liquid cloud suggests that SPL is essentially a cloud chamber that produces a consistent cloud under moist, westerly flow during the winter. A variable Cloud Condensation Nuclei (CCN) related inverse relationship between CDNC and NMD strengthened when the data were stratified by LWC. Some of this variation is due to changes in cloud base height below SPL. While there was a weak inverse correlation between LWC and IWC in the data as a whole, a stronger relationship was demonstrated for a case study on February 9, 2014 during IFRACS. A minimum LWC of 0.05 g m$^{-3}$ showed that the cloud was not completely glaciated on this day. Erosion of the droplet distribution at high IWC was attributed to the Wegener-Bergeron-Findeisen process although the high IWC was caused by a 10 fold increase in Ni. A relationship found between large cloud droplet concentration (25-35 µm) and small ice particles (75-200 µm) under cold (<-12 °C) and warm (>-8 °C) conditions suggests ice particle production by contact or immersion freezing. Such a relationship under warm conditions could be indicative of biological ice nuclei. There was no direct evidence of secondary ice production. The effect of blowing snow was evaluated by comparing the relative (percent) ice particle PSDs at high and low wind speeds. These were similar, contrary to expectation for blowing snow. However, correlation between wind speed and ice crystal concentration may support this explanation for high crystal concentrations at the surface. Further experimental work is needed to resolve this issue.

## 1 Introduction

Aerosols and their effects on cloud microphysical properties have been shown to alter precipitation formation and distribution over complex terrain (e.g., Pruppacher and Klett, 1997; Borys et al., 2003; Rosenfeld and Givati, 2006; Lowenthal et al., 2011; Saleeby et al., 2013). Higher concentrations of cloud condensation nuclei (CCN) produce more



numerous but smaller cloud droplets (Twomey et al., 1984; Pen1g et al., 2002; Lowenthal et al., 2002). This leads to decreased riming efficiency and a decrease precipitation on the windward slope (Borys et al., 2000, 2003) and has been shown to redistribute precipitation over mountain barriers in modeling studies (Saleeby et al., 2009, 2013).

    There are numerous studies and reviews of ice nucleation theory, measurements and modeling (Vali, 1996, 1999; Diehl et al., 2006; Hoose and Möhler, 2012; Moreno et al., 2013; Murray et al., 2012; Knopf and Alpert, 2013; Kanji
et al., 2017; Knopf et al., 2018). In MPC, a small fraction of aerosols can act as heterogeneous ice nucleating particles (INP) and produce ice through four known freezing modes: deposition, immersion, condensation, and contact freezing. Deposition freezing occurs when water vapor deposits directly onto an INP and generally occurs at temperatures below other heterogeneous freezing modes (Lohmann and Diehl, 2006). Contact freezing occurs when an INP comes into contact with a supercooled cloud droplet and initiates freezing. Immersion freezing and condensation freezing are
more difficult to differentiate. Immersion freezing occurs when a supercooled cloud droplet forms, grows on an INP, and then freezes. Condensation freezing occurs when water vapor condenses onto an INP and forms a haze a droplet which freezes.

    However, recent work in Arctic MPC showed that ice was observed only after a liquid cloud layer had developed and found no evidence of condensation freezing (de Boer et al. 2011). The relative efficiencies of contact and
immersion nucleation are discussed by Pitter and Pruppacher (1973), Lohmann and Diehl (2006) and Nagare et al. (2016). Contact freezing has been found to occur at higher temperatures than immersion freezing for a given INP (Pitter and Pruppacher 1973; Lohmann and Diehl, 2006; Nagare et al., 2016). Biological INP have been found to produce ice at relatively higher temperatures than non biological INP (Levin and Yankofsky, 1964; Du et al., 2017).

    Secondary ice production (SIP) processes are reviewed by Field et al. (2017). Sullivan et al. (2018) modeled SIP
by rime splintering (Hallett-Mossop process), droplet shattering, and collisional breakup with ice particle enhancement depending on temperature, updraft velocity, and INP concentration. Rime splintering is thought to occur when a supercooled droplet with a diameter larger than ~25 μm freezes onto an ice particle or other surface and shatters at temperatures between -8 and -3 °C (Hallett and Mossop, 1974; Mossop, 1985). Keppas et al. (2017) found evidence for rime splintering in warm (-6 to 0°C) frontal clouds. Here, "lolly pop" shaped crystals formed by riming of columnar
crystals by droplets larger than 100 μm were associated with high concentrations of small columnar crystals. Rangno and Hobbs (2001) concluded that shattering of freezing droplets (>50 μm) could have accounted for high observed ice particle concentrations in Arctic stratus.

    At mountaintop observatories, ice crystal concentrations frequently exceed aircraft measurements by an order of magnitude or more (Roger and Vali, 1987; Geerts et al., 2015; Lloyd et al., 2015; Beck et al., 2017). Lloyd et al. (2015)
considered blowing snow, rime splintering, and detachment of surface frost (Bacon et al., 1998) as sources of high ice particle concentrations at the Jungraujoch Sphinx Observatory (JFJ). They ultimately favored the latter mechanism by process of elimination, albeit with no direct evidence. In contrast, Beck et al. (2017) suggested that the enhanced ice crystal concentrations at mountaintops are due to blowing snow, turbulence near the mountain surface, or convergence of ice crystals near mountaintop due to orographic lifting.

Several studies have shown a link between cloud droplet size and ice particle concentrations (e.g. Hobbs and Rangno, 1985; Rangno and Hobbs, 2001; Lance et al. 2011; de Boer et al., 2011). Hobbs and Rangno (1985) found a



strong relationship between the width of cloud droplet spectra and ice particle concentrations in cumuliform and stratiform clouds where cloud top temperature ranged between -36 and -6°C. Lance et al. (2011) found higher concentrations of ice particles larger than 400 μm in clean Arctic clouds with larger droplets sizes than in polluted

Arctic clouds with smaller but more numerous drops.

The aforementioned studies have furthered the understanding of precipitation processes and distributions in complex terrain from dynamical and microphysical perspectives but due to the lack of data, were unable to establish a link between the cloud microphysics aloft and the surface. Rogers and Vali (1987) observed cloud microphysics at both the Elk Mountain Observatory (EMO) located in the Medicine Bow Mountains of Southern Wyoming and from

the University of Wyoming Queen Air (UWQA) aircraft. Comparisons between crystal concentrations at EMO and on the UWQA routinely showed higher crystal concentrations at the surface. The authors attributed the higher surface concentrations to an unspecified process of ice crystal generation in supercooled orographic clouds in contact with snow covered mountain surfaces. However, blowing snow can also introduce the potential for artifacts in observed ice crystal concentrations at mountaintop locations (Roger and Vali, 1987; Geerts et al., 2015).

The Storm Peak Lab Cloud Property Validation Experiment (StormVEx) was conducted from November 15, 2010 to April 25, 2011 at DRI's Storm Peak Laboratory (SPL) to produce a correlative data set to validate cloud retrievals using in situ measurements at SPL (Mace et al., 2010; Matrosov et al., 2012). The Isotopic Fractionation in Snow (IFRACS) study was conducted at SPL from January 20 to February 27, 2014 to explore the impacts of microphysical processes in wintertime orographic clouds on the water isotopic composition of falling snow

(Lowenthal et al., 2016; Moore et al., 2016). This paper examines microphysical properties of wintertime orographic MPC at SPL using data collected during StormVEx and IFRACS. A large record of concurrent measurements of ice and supercooled liquid water was generated. These data enable exploration of relationships among microphysical properties in a statistical sense, the temporal variation of cloud properties over a 3 year period at this site, the relationship between the ice and liquid phases, and ice formation mechanisms. Potential measurement artifacts due to

instrumental characteristics and blowing snow are evaluated.

## 2 Methods

Storm Peak Laboratory (SPL, 3210 m ASL; 40.456570°N, 106.739948°W) is located on the summit of Mt. Werner in the Park Range near Steamboat Springs, Colorado (Wetzel et al., 2004). In wintertime, SPL is in snowing, supercooled liquid cloud roughly 25% of the time (Borys and Wetzel, 1997). Snow storms occur roughly weekly

under a variety of synoptic conditions (Rauber and Grant, 1986; Rauber et al., 1986; Borys and Wetzel, 1997). As noted by Lowenthal et al. (2016), given sufficient moisture during winter, a cloud forms and produces persistent snowfall at SPL. Winds are generally from the west or northwest during snowfall events. Cloud and snowfall can be inhibited due to blocking by the Flat Top Range (maximum elevation 3768 m ASL) under flow from the southwest.

Cloud microphysical measurements were made using the same instruments during StormVEx and IFRACS.

Cloud droplet number concentrations (CDNC) and particle size distributions (PSDs) from 2-47 μm were measured with an aspirated Droplet Measurement Technologies, Inc. (Boulder, CO) SPP-100 forward scattering spectrometer probe. Liquid water content was calculated from the SPP-100 PSDs. During IFRACS, the SPP-100 inlet was equipped



with a "scarf tube", which narrows and accelerates the flow in the sample volume to 25 m s$^{-1}$ according to the manufacturer. The face velocity at the center of the inlet was measured at 9.4 m s$^{-1}$, which corresponds to a velocity

of 26.7 m s$^{-1}$ in the sample volume. The scarf tube was removed during StormVEx such that the face velocity at the inlet should have been the same as that in the sample volume. There were attempts to measure the face velocity during StormVEx but these were inconsistent. Therefore, StormVEx SPP-100 concentrations were recalculated using the face velocity of 9.4 m s$^{-1}$ measured during IFRACS.

Ice particle PSDs were measured with a DMT CIP (Cloud Imaging Probe [25-1600 μm]) optical array probe (OAP)

with 64 channels and a resolution of 25 μm. The cloud probes were calibrated and serviced prior to each field campaign. During IFRACS, an Applied Technologies, Inc. (Longmont, CO) SATI 3 axis sonic anemometer supplied the wind speed along the horizontal axis of the CIP probe. For aircraft measurements, this is referred to as true air speed (TAS). During StormVEx, a Lufft Ventus UMB 2 axis sonic anemometer was substituted for the Applied Technologies, Inc. instrument after February 8, 2011. Data were collected at 1 Hz. The cloud probes and sonic

anemometers were mounted on a rotating wind vane (to orient them into the wind) located on the west (upwind) railing of the roof approximately 6 m above the snow surface.

The 2 D CIP images from StormVEx and IFRACS were processed using the Optical Array Shadow Imaging Software (OASIS) program developed at the University of Manchester (Crosier et al., 2011; Lloyd et al., 2015) and marketed by DMT (http://www.dropletmeasurement.com/optical-array-shadow-imaging-software-oasis). The CIP

depth of field was corrected as a function of particle size (Baumgardner and Korolev, 1997). Ice particle shattering on the probe tips was found to be insignificant based on particle interarrival time (Field et al., 2006). This is consistent with relatively low wind speed at the surface compared with aircraft speeds (~100 m s$^{1}$). Concentrations in the first two CIP channels (<62.5 μm) were ignored because of sizing uncertainties (Korolev et al., 1998: Strapp et al., 2001) and because some of these particles are likely to be cloud droplets in mixed phase clouds (MPC). The total CIP

concentration excluding the first two channels is referred to as Ni. The center in approach, which includes particles that obscure an end diode, was used to identify particles and estimate the sample volume (Heymsfield and Parrish, 1978). Particle size was described as the area equivalent diameter, i.e., the diameter of a circle with the same area as the particle, as determined from the number of shadowed pixels and the probe resolution. Ice water content (IWC) was estimated by OASIS using the approach of Brown and Francis (1995). This estimate is uncertain because mass

dimensional relationships vary significantly with ice particle habit, riming extent, aggregation, and temperature (Mitchell, 1996; Schmitt and Heymsfield, 2010).

In aircraft studies, the volume of air sampled by cloud probes is related to TAS. At aircraft speeds, particles are sampled along the horizontal axes of and perpendicular to the sample area of the cloud probes. This is not necessarily the case with ground based sampling, even when the probes are mounted on a wind vane such as those used at SPL or

JFJ, where cloud probes were mechanically oriented into the wind based on sonic anemometer measurements (Lloyd et al., 2015). If the particle trajectory is not as described above, the particles can appear misshapen but not necessarily miss sized according to the area equivalent diameter. CIP data used in the following analysis were constrained as follows: 1) 1 second TAS >1 and <20 m s$^{-1}$. A lower limit is needed to ensure that particles traverse the CIP array as close to horizontally as possible. Note that the updraft near the mountain tends to impart a horizontal trajectory on





falling ice particles (Borys et al., 2000). An upper limit is needed to guard against contamination by blowing snow. During StormVEx and IFRACS, snow and supercooled cloud water samples were collected in bags and on cloud sieves (Borys et al., 2000). Such sampling is not practical at wind speeds above 15 m s$^{-1}$, where snow may blow out of the bags and the cloud sieves may become overloaded. For the January and February period during StormVEx, TAS was >20 m s$^{-1}$ during only 34/492995 (0.007%) of seconds when the CIP measured particles. The corresponding

frequency during IFRACS was 3663/338230 (1.1%). Five minute average temperature, pressure, and humidity were measured by the SPL weather station. Water vapor concentration and isotopic composition were measured during IFRACS with a Picarro L2130-i water vapor isotopic analyzer (Lowenthal et al., 2016).

## 3 Results and Discussion

The full StormVEx program lasted nearly 6 months, from November, 2010 through April, 2011, while IFRACS was

designed as a 6 week field project in January and February, 2014. During IFRACS, the Picarro began collecting data on January 20, however, the weather was clear until January 27 (Lowenthal et al., 2016). For a consistent comparison between the two studies, StormVEx data are limited to January and February, 2011. Cloud probe measurements were made on 30 days during StormVEx and 15 days during IFRACS. Measurement periods during StormVEx were intended for comparison with ground based remote sensing instruments. The probes were turned on when it started

snowing but were not necessarily turned off if SPL was not in MPC. Measurements during IFRACS were started only when SPL was in MPC to sample liquid and ice for isotopic analysis. While there were twice as many sampling days during StormVEx, the CIP probe measured particles for 101.4 and 77.2 hours during StormVEx and IFRACS, respectively. The 1 second data were averaged to 1 minute with a 75% (at least 45 seconds) data completeness requirement. To ensure that the measurements represented MPC, only seconds when Ni was >0, LWC was >0.01 g

m$^{-3}$ and CDNC was >10 cm$^{-3}$, were included. With these constraints, there were 49.2 and 43 hours of concurrent MPC measurements during StormVEX and IFRACS, respectively.

### 3.1 SPP-100 and CIP Particle Size Distributions

Average PSDs calculated from concurrent 1 minute average SPP-100 and CIP measurements are shown in Figs. 1a and 1b for StormVEx and IFRACS, respectively. The average PSDs were similar in the two studies. Corresponding

averages of 1 minute CIP and SPP-100 concentrations are summarized in Table 1. Table 1 shows that LWC and CDNC were similar in the two studies, although average IWC during IFRACS was twice that during StormVEx. Small (75-200 μm, referred to as Conc75-200) and large (>400 μm) ice particle concentrations were also higher during IFRACS. The average LWC at SPL was more than an order of magnitude lower than LWC observed in the Sierra Nevada (1.5 g m$^{-3}$) and Cascade (2 g m$^{-3}$) mountains, respectively (Lamb et al., 1976; Hobbs, 1975). The ratios of average Conc75-

200 to average Ni were 91 and 83% during StormVEx and IFRACS, respectively. Based on their coefficients of variation, liquid cloud properties (CDNC and LWC) were much less variable than Conc75-200, large ice particles, and Ni at SPL.

     While the first CIP channel, nominally 12.5-37.5 μm, lines up with the SPP-100 PSD at ~25 μm in both studies (Fig. 1), concentrations of SPP-100 particles larger than 25 μm undershot the CIP PSD during StormVEx but not



IFRACS. Concentrations of droplets larger than 25 µm were significantly lower during StormVEx. The average TAS was 6.1 m s$^{-1}$ during StormVEx and 6.0 m s$^{-1}$ during IFRACS. At an SPP-100 sampling flow rate of 9.4 m s$^{-1}$ and an average TAS of ~6 m s$^{-1}$, sampling is anisokinetic, leading to oversampling of smaller droplets. Ideally, the flow through the SPP-100 sample tube should increase as the square of the radius inside the scarf tube. However, the behavior of the flow at the leading edge of the scarf tube could be turbulent, resulting in entrainment of larger particles.

This is consistent with the PSDs shown in Fig. 1 and the slightly lower LWC and mean diameter (NMD) during StormVEx (Table 1).

Spherical liquid drops and irregular ice particles can be distinguished with image analysis, however, this is only possible for particles with area equivalent diameters larger than about 110 µm for the CIP (Crosier et al., 2011). To determine whether the CIP measured liquid droplets in MPC, the average of the 1 second CIP PSDs in mixed phase

(wet) cases were compared with dry cases when Conc75-200 was >0 and LWC was zero [no particles detected by the SPP-100]). Figure 2 shows the ratio of the average of 1 second wet to average dry CIP concentrations as a function of size for StormVEx and IFRACS. In both studies, the ratio was elevated in the first CIP channel only. The ratio decreased significantly and was flat between the third and eighth CIP channels, i.e., Conc75-200. Thus, the CIP measurements were affected by cloud droplets only in the first CIP channel.

Average Conc75-200 was higher under wet than dry conditions: 78 versus 49 L$^{-1}$ during StormVEx and 118 versus 21 L$^{-1}$ during IFRACS. This could be an indication of liquid mediated ice production (Rangno and Hobbs, 2001; de Boer et al., 2011; Lance et al., 2011). Note that average TAS under wet and dry conditions were similar, i.e., 5.9 and 6.5 m s$^{-1}$, respectively, during StormVEx and 5.9 and 5.2 m s$^{-1}$, respectively, during IFRACS. There is an apparent inconsistency in this analysis, which is that when particles are counted in the first CIP channel under "dry"

conditions, they should also be measured by the SPP-100. The average concentrations in the first CIP channel under "dry" conditions were 0.32 and 0.06 cm$^{-3}$ during StormVEx and IFRACS, respectively. These values can be inaccurate because of sizing uncertainty in the first two CIP channels. The actual concentration of particles in CIP channel 1 under "dry" conditions may also be below the limit of detection of the SPP-100. The impact of ice particles on SPP-100 measurements cannot be observed directly with these instruments. Taken at face value, the magnitude of the ratio

of wet/dry concentrations in CIP channel 1 places an upper limit on the effect of ice particles on the SPP-100 measurements. On average, droplets were 34 times more abundant than crystals in the 12.5-37.5 µm size range while the corresponding ratio during StormVex was only 3.7.

The distributions of Conc75-200, wind speed and temperature as a function of wind direction during StormVEx and IFRACS are summarized in Table 2. Winds were from the NW sector ~75.3 and 57% of the time during StormVEx

and IFRACS, respectively. There was one 5 minute period during IFRACS when the wind direction was 11 degrees. High Conc75-200 were seen in the NW sector in both studies but the highest concentrations were seen in the NNW sector, albeit at low frequency. When segregated by wind direction, there was no relationship between Conc75-200 and temperature or wind speed in either study.

**3.2 Supercooled Liquid Cloud Microphysics**





In non precipitating warm clouds, an increase in CCN should increase CDNC while decreasing droplet size at constant
      LWC (Albrecht, 1989). Smaller drops may inhibit collision coalescence and precipitation and increase LWC (Zheng
      et al., 2010). Borys et al. (2000) demonstrated a direct relationship between clear air equivalent sulfate concentration
      (a surrogate for pre cloud CCN) and CDNC and an inverse relationship between CDNC and droplet size (NMD) in
      MPC at SPL. In such clouds, the droplet distribution may be impacted by riming of ice particles and by transitions

between the liquid and ice phases. Figure 3 presents the relationship between 1 minute droplet NMD and CDNC in
      MPC during StormVEx (Fig. 3a) and IFRACS (Fig. 3c). The relationship is stronger when the data are stratified by
      LWC. The average NMD and CDNC were calculated for each of the four ranges of LWC in Fig. 3 and are plotted in
      the figures as a function of LWC. NMD and CDNC increased monotonically with LWC in both studies. This is
      consistent with enhanced growth of droplets as cloud base drops below SPL. However, for CDNC to increase with

LWC, either the supersaturation must increase or CCN aerosols must become entrained in the cloud between cloud
      base and SPL. Figures 3b and 3d present average SPP-100 PSDs for low (0.05-0.1 g m$^{-3}$) and high (0.2-0.3 g m$^{-3}$)
      LWC, corresponding to Figs. 3a and 3c, respectively. The distributions are shifted to larger sizes at high LWC and the
      increase in CDNC is evident for droplet sizes larger than 10 µm. Note that the shift in the PSDs to larger sizes at high
      LWC stops at about 35 µm, i.e., the concentration of very large drops is higher at low LWC. This could indicate a

preferential loss of very large drops to riming at high LWC.

### 3.3 Relationship between LWC and IWC

      As noted above with respect to Table 1, liquid cloud microphysical properties at SPL were less variable than those of
      the ice phase. One reason for this is that the ice phase is impacted by processes occurring upwind and at higher altitude.
      Lowenthal et al. (2011; 2016) estimated that most of the snow mass was formed within 1 km above SPL. This does

not preclude ice nucleation at higher altitudes, as small, freshly nucleated crystals contribute little to IWC. Even though
      riming occurs, most efficiently for large droplets, it is not apparent from Figs. 1 and 4 that the liquid cloud was
      impacted by the ice phase. Indeed, the Pearson and Spearman Rank (non parametric) correlations between all
      concurrent 1 minute average IWC and LWC were only -0.18 and -0.10, respectively, during StormVEx and -0.13 and
      -0.16, respectively, during IFRACS. The effect of outliers, characteristic of skewed distributions, is reduced with the

non parametric statistic. Henceforth, the Spearman Rank correlation is displayed in parenthesis after the Pearson
      correlation. Scatter plots of IWC versus LWC (not shown) resemble a solid cluster of points filling a right triangle.
      The edge (hypotenuse) in the data suggests that there were periods when IWC and LWC were more strongly
      anticorrelated. If only days with at least 2 hours of valid, 1 minute average data are considered, there were 4 out of 11
      and 3 out of 11 days during StormVex and IFRACS, respectively, where the Pearson and Spearman Rank correlations

between IWC and LWC were less than -0.5.

      A sampling day during IFRACS with relatively high average IWC (0.23 g m$^{-3}$) and LWC (0.182 g m$^{-3}$) was
      identified for closer examination. Figure 4 presents time series of 1 minute average IWC and LWC on February 9,
      2014. In this case, the correlation between IWC and LWC was -0.59 (-0.60), suggesting interaction between the ice
      and liquid phases. The minimum LWC was 0.05 g m$^{-3}$, i.e., the cloud at SPL was never fully glaciated on this day. To

contrast periods with high and low IWC, average SPP-100 PSDs were calculated for the period between 12:45 and



13:17 MST (Fig. 4) and for minutes outside of that interval with the additional constraint that the LWC/IWC ratio was greater than 2. These PSDs are presented in Fig. 5. The average IWC and LWC were 0.72 and 0.088 and 0.054 and 0.25 g m$^{-3}$ for the high and low IWC periods respectively. The average IWC and LWC during the high IWC and high LWC periods were 3.7 and 1.98 times higher, respectively, than the study wide averages (Table 1). Compared with the low IWC period, the high IWC SPP-100 PSD displays a marked loss of particles with diameters between ~5 and 23 μm. The corresponding loss of liquid water was 0.181 g m$^{-3}$ (Fig. 5). The most obvious explanation is evaporation of droplets (Wegener-Bergeron-Findeisen process). The loss of LWC is much lower than the more than order of magnitude difference in IWC for the two cases.

The increase in IWC is due to an order of magnitude higher Ni concentration at high IWC (525 L$^{-1}$) compared to low IWC (50 L$^{-1}$). The correlation between IWC and Ni was 0.98 (0.98). At the same time, the concentration of large droplets (25-35 μm, CDNC25-35) was higher at high IWC (976 L$^{-1}$) than at low IWC (422 L$^{-1}$). This could imply a link between ice production and the presence of large cloud droplets (Hobbs and Rangno, 1985; de Boer et al., 2011; Lance et al., 2011).

There were no relationships between LWC or IWC and either temperature or water vapor concentration, which were relatively invariant, i.e., -5.4±0.3 °C and 8064±204 ppmv, respectively. The correlations between TAS and MTAS (maximum 1 second TAS) and IWC were 0.46 (0.42) and 0.66 (0.60), respectively. The correlations between TAS and MTAS and Ni were 0.42 (0.38) and 0.62 (0.56), respectively. The higher correlations for MTAS suggest that the high IWC (Ni) period could have been influenced by blowing. Average TAS and MTAS during the high and low IWC periods were 8.3 and 16.6 and 5.6 and 8.9 m s$^{-1}$, respectively. A difference of 2.7 m s$^{-1}$ in TAS between the high and low IWC periods is unlikely to have caused the 10 fold increase in Ni during the high IWC period. However, the near doubling of average MTAS to ~17 m s$^{-1}$ during the high IWC period suggests that the large increase in Ni could have been related to blowing snow.

### 3.4 Liquid Mediated Ice Production

In this section, the hypothesis that ice production in MPC at SPL is related to large droplet concentration is examined (Hobbs and Rangno, 1985; Lance et al., 2011). To reiterate, Fig. 2 demonstrates that the CIP measured cloud droplets in the first but not higher channels. Noting that particles were measured by the CIP in channel 1 under dry conditions, the ratio of wet/dry concentration in CIP channel 1 constrains the effect of ice particles on the SPP-100 measurements. During StormVEx, a wet/dry a ratio of ~4 suggests that ~20% of CDNC25-35 could have been ice particles. During IFRACS, a wet/dry ratio of 34 suggests that the effect of ice particles on CDNC25-35 was negligible.

The relationships between 1 minute average CDNC25-35 and Conc75-200 were examined under cold (<-12 °C) and warm (>-8 °C) conditions. This is intended to distinguish cold and warm primary or secondary ice production processes. Figures 6a and 6c present relationships for StormVEx under cold and warm conditions, respectively and Figs. 6b and 7d present the corresponding relationships for IFRACS. The average temperatures for all 1 minute averages (Table 1) during StormVEx and IFRACS were -12.8±2.8 and -8.2±3.6 °C, respectively. Figure 6a shows a moderate relationship (r=0.57 [0.45]) between CDNC25-35 and Conc75-200 at cold temperature during StormVEx but no direct relationship at warm temperature (r=-0.10 [-0.45]). Note that this is based on relatively few (29) data



points. During IFRACS (Fig. 6b), there was a strong relationship (r=0.90 [0.88]) at cold temperature but a weaker one (r=0.42 [0.53]) at warm temperature (Fig. 6d). Figure 6d suggests that the warm data points followed two trends, one similar to the cold points in Fig. 6b and the other similar to the flat distribution of warm points in Fig. 6c.

Temperatures at SPL during StormVEx and IFRACS were less than -15°C only 9% of the time and thus small crystals could not have nucleated homogeneously or by deposition. Given the relationships between large droplet and small ice crystal concentrations, is the temperature range at SPL consistent with immersion and/or contact freezing? This appears to be the case at colder temperatures (<-12 °C) at SPL for contact freezing, as seen in Figs. 7 and 13 in Moreno et al. (2013) and for immersion freezing, particularly for biological INP (Levin and Yankofsky, 1964; Du et

al., 2017; Kanji et al., 2017). The large droplet – small crystal relationship was considerably noisier during StormVEx (Fig. 6a) than IFRACS (Figure 6c). This could reflect contamination of CDNC25-35 by ice particles or a problem with the measurement of large droplets during StormVEx (discussed above). The relationship is ambiguous at warmer temperatures. Temperatures were >-8 °C on 3 out of 30 sampling days (2.7% of sample minutes) during StormVEx and 9 out of 15 sampling days (54% of sample minutes) during IFRACS. Assuming that biological INP are more

prevalent under warmer conditions (Stopelli et al., 2015), the direct relationship during the warm IFRACS sampling periods is consistent with contact or immersion freezing involving biological INP. Bowers et al. (2009) observed biological INP at SPL which froze at temperatures ≥ 10 °C in the immersion mode.

Secondary ice production (SIP) mechanisms have been extensively reviewed (e.g., Field et al., 2017). Sullivan et al. (2018) modeled SIP by rime splintering, droplet shattering, and collisional breakup. Rangno and Hobbs (2001)

concluded that shattering of large droplets (>50 μm) upon freezing could have accounted for high observed ice particle concentrations in Arctic stratus. While there is no evidence of droplets this large at SPL, they could be present upwind and above SPL. Keppas et al. (2017) concluded that rime splintering occurred in warm (-6 to 0 °C) frontal clouds. "Lolly pop" shaped crystals were taken as evidence of riming of columnar crystals by droplets larger than 100 μm. Neither "lolly pops" nor droplets this large have been observed in MPC at SPL. Lloyd et al. (2015) considered blowing

snow, rime splintering, and detachment of surface frost (Bacon et al., 1998) as sources of high ice particle concentrations at JFJ. They ultimately favored the latter process, albeit with no direct evidence. There is also no evidence regarding surface frost splinters at SPL. Rime splintering could have been responsible for the relationship between CDNC25-35 and Conc75-200 during IFRACS under warm conditions although it is not clear why this wouldn't also have been the case during StormVEx. Perhaps there were too few warm 1 minute periods during

StormVEx to establish a meaningful relationship. For rime splintering to account for a relationship between CDNC25-35 and Conc75-200, the rime mass fraction as well as the number of splinters produced by each rimed droplet would have to be consistent. This is farfetched.

### 3.5 Blowing Snow

Blowing snow can cause significant artifacts in ice crystal measurements at surface locations. Rogers and Vali (1987)

found higher ice crystal concentrations at the Elk Mountain Observatory compared with those observed aloft on the University of Wyoming Queen air but discounted blowing snow as the explanation for this difference. Lloyd et al. (2015) concluded that high ice crystal concentrations at JFJ were not caused by blowing snow. Geerts et al. (2015)





compared CIP concentrations ($\geq$75 µm) at SPL with those measured aboard the University of Wyoming King Air (UWKA) during the Colorado Airborne Multiphase Cloud Study (CAMPS) when the aircraft was in the vicinity of
SPL. Concentrations were considerably higher at SPL when the maximum wind speed associated with 5 minute average measurements was above about 4 m s$^{-1}$. This was attributed to blowing snow. However, a valid comparison between aircraft and surface measurements depends on the assumption that both platforms measure the same ice crystal population. This would require establishing crystal trajectories from a point upwind aloft to a point downwind at the surface. Even if a direct link between the PSDs aloft and at the surface could be demonstrated, the falling crystal
PSD is likely to be modified by depositional growth at ice supersaturation in the low level liquid cloud, riming and aggregation, or sublimation in subsaturated regions. The ice crystal PSD measured at the surface can also be enhanced by ice production near the surface, as discussed above, secondary ice production, or blowing snow.

Beck et al, (2017) conducted ice particle measurements at various heights on a 10 m tower at the Sonnblick Observatory (SBO) in Rauris, Austria. They suggested that during cloud free periods, a rapid decrease in ice crystal
concentration with height could be explained by blowing snow. However, the wind speeds during those periods were around 17 m s$^{-1}$, which is significantly higher than the 1 minute average wind speeds in the SPL analysis. In contrast, when SBO was in liquid cloud, a consistent decrease in ice crystal concentration with height on the tower was not observed. Rather, varying vertical profiles of crystal concentration were attributed to advection of ice crystals in a turbulent layer above the snow surface or enhancement of ice crystal concentration due to a convergence zone of ice
crystals at the mountaintop associated with orographic flow over the barrier.

Correlations between 1 minute average TAS and Conc75-200 during StormVEx and IFRACS were 0.38 (0.36) and 0.54 (0.47), respectively. These moderate correlations could be taken as evidence for blowing snow. Note that the corresponding correlations in the 1 second data were lower, i.e., 0.23 (0.25) and 0.14 (0.22), respectively. The lower correlations in the 1 second data suggest that if surface snow had been resuspended by the wind, it would have occurred
at some distance upwind of the building. Table 3 presents average Conc75-200 over ranges of TAS and CDNC25-35 during StormVEx and IFRACS. Conc75-200 increases monotonically, if not linearly, with both TAS and CDNC25-35 in both studies. Finally, relationships between Conc75-200 and TAS, CDNC25-35, and temperature were examined using stepwise and non parametric regression. The results are shown in Table 4. The higher a variable's contribution (partial r square) to the model r square, the greater its importance as an explanatory variable. For StormVEx, TAS had
the highest partial r square, followed by CDNC25-35 and Temp, whose contribution was negligible. However, for IFRACS, CDNC25-35 was the primary contributor to r square, followed by TAS and Temp (also negligible). The relatively smaller contribution of CDNC25-35 to the variance of Conc75-200 during StormVEx could be due to a measurement problem, as discussed above. Indeed, average CDNC25-35 concentrations during StormVEx and IFRACS were 47 and 231 L$^{-1}$, respectively, while total CDNC were similar (Table 1).

Smaller crystals should be more efficiently lofted from the snow surface and remain suspended farther downwind than larger ones (Schmidt, 1982). Thus, blowing snow should result in a relative enrichment of small crystals in the CIP PSD, independent of absolute concentration. Average 1 minute CIP PSDs were calculated, normalized to average Ni, and expressed as percentages. These are presented for high (8-12 m s$^{-1}$) and low (1-4 m s$^{-1}$) TAS in Figure 7. Contrary to expectation for blowing snow, there is little difference between the relative PSDs at low and high TAS up





to approximately 200 and 300 μm during StormVEx and IFRACS. The cumulative increase in small particles and corresponding decrease in large particles at the point where the high and low TAS PSDs cross was 7% in both studies. This small enhancement of small particles at high TAS cannot explain the large differences between surface and aircraft measurements observed by Rogers and Vali (1987) and Geerts et al. (2015). However, it could be due to an equivalent loss of large particles through collisional breakup. Increased turbulence should enhance collisional breakup

of crystals by increasing collision frequency (Vardiman, 1978; Lohmann et al., 2016). Combining data from both studies, the correlations between u (TAS) and $\sigma_u$ and between $w$ and $\sigma_w$ were 0.76 (0.75) and 0.71 (0.73), respectively, i.e., there was more turbulence at higher TAS.

## 4 Conclusions

Studies of mixed phased orographic clouds (MPC) were conducted at the Storm Peak Laboratory (SPL) in

northwestern Colorado in January and February during StormVEx (2011) and IFRACS (2014). In total, the data represent ~92 hours when SPL was immersed in supercooled liquid cloud and it was snowing. On average, liquid cloud PSDs, CDNC, NMD, and LWC were similar between years while Ni and IWC were 48 and 114% higher, respectively, during IFRACS. Average wind speeds were similar (~ 6 m s$^{-1}$) in both studies while average temperatures were colder during StormVEx (-12.8 °C) than IFRACS (-8.2 °C). Supercooled liquid cloud properties at SPL were

consistent between the two studies. The microphysical properties of ice particles were more variable as they depend on the structure of the cloud above and upstream of SPL.

The inverse relationship between cloud droplet size (NMD) and concentration (CDNC) is related to CCN at SPL (Borys et al., 2000). This relationship is stronger when the data are stratified by LWC. Both CDNC and NMD increase with increasing LWC, demonstrating droplet growth and enhanced activation of or entrainment of CCN below SPL.

Future studies at SPL would benefit from direct measurement of cloud base height. There was a weak relationship between LWC and IWC for all data (the correlation was -0.18 (-0.10) and -0.13 (-0.16) during StormVEx and IFRACS, respectively), however, a stronger inverse relationship was evident on several days during each study. This was demonstrated for a case on February 9, 2014, where the correlation between IWC and LWC was -0.59 (-0.60). During a period of maximum IWC on this day, the droplet PSD showed a significant loss of liquid water and a decrease in

droplet concentration compared to periods with low IWC and high LWC. As there was an order of magnitude increase in the ice crystal concentration (Ni) between the high and low IWC periods, the loss of LWC was likely due to crystal growth at the expense of evaporating droplets (Wegener-Bergeron-Findeisen process).

Relationships between large cloud droplets (CDNC25-35) and small ice crystals (Conc75-200) suggest that droplet freezing (contact or immersion) was involved in ice production at SPL. This relationship was evident during

StormVEx only at temperatures below -12 °C. During IFRACS, relationships were seen under cold (<-12 °C) and warm (>-8 °C) conditions. Warmer temperatures during IFRACS could have been associated with an increase in biological aerosols which have been shown to be effective immersion and contact INP at warmer temperatures than inorganic INP. There is no evidence that secondary ice production mechanisms such as rime splintering, large droplet freezing, or frost splintering influenced Conc75-200 at SPL. It is unclear how these processes could have produced

the observed correlations between large droplet and small ice crystal concentrations.



Blowing snow can significantly impact ice crystal concentrations at the surface and has been invoked to explain differences between surface and aircraft ice crystal measurements. The effect of blowing snow on ice crystal measurements at SPL was evaluated based on the assumption that blowing should preferentially enhance the relative abundance of small crystals in the ice crystal PSD. Comparison of the relative (expressed as percentages of the total) ice crystal PSDs at high (8-12 m s$^{-1}$) and low (1-4 m s$^{-1}$) TAS cannot explain previously reported orders of magnitude differences between surface and aircraft measurements. While the temperature dependent relationships between CDNC25-35 and Conc75-200 suggest ice production by droplet freezing, moderate correlations and a statistical dependence of Conc75-200 on TAS can be taken as evidence of blowing snow. It is possible that both primary production and blowing snow were active at SPL. These results highlight the need for targeted experiments to quantify the contributions of blowing snow to ice crystal concentrations at mountaintop locations.

### Data Availability

Data are available at: https://www.dri.edu/doug-lowenthal-research-reviews

### Author Contribution

DL was a principal investigator on IFRACS. AGH is the director of the Desert Research Institute's Storm Peak Laboratory. AGH and GM were principal investigators on StormVEx. RD was a graduate student at DRI who worked on the IFRACS field experiment and used the results in his Master's thesis. IM is the site manager at Storm Peak Laboratory. RB is Professor Emeritus at DRI and worked on the IFRACS field experiment.

### Competing Interests

The authors declare that they have no conflict of interest.

### Acknowledgements

This work was supported by Department of Energy Atmospheric System Research Program grant DE-SC0014304 and by National Science Foundation Division of Atmospheric Sciences grant AGS-1260462. Logistical assistance from the Steamboat Ski and Resort Corporation is greatly appreciated. The Desert Research is an equal opportunity service provider and employer and is a permittee of the Medicine-Bow and Routt National Forests. We would especially like to thank and acknowledge the hard work of many people who made the StormVEx project possible, including the many DOE ATSC and ASR staff, Storm Peak Laboratory (SPL) local volunteers, the Steamboat Ski and Resort Corporation, the U.S. Forest Service, the Grand Junction National Weather Service office, and all of the graduate students (Betsy Berry, Stewart Evans, Ben Hillman, Will Mace, Clint Schmidt, Carolyn Stwertka, Adam Varble, and Christy Wall), who put considerable effort into data collection.




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



**Table 1**. Average of concurrent 1 minute CIP and SPP-100 measurements during StormVEx and IFRACS. The values in parentheses are the coefficients of variation.

| | CIP | | | | | SPP-100 | | | | | | |
| | Conc75-200[a] (L$^{-1}$) | Large[b] (L$^{-1}$) | Ni[c] | Conc75-200/ Ni (%) | Large/ Ni (%) | IWC[d] (g m$^{-3}$) | CDNC[e] (cm$^{-3}$) | LWC[f] (g m$^{-3}$) | NMD[g] (μm) | TAS (m s$^{-1}$) | Temp. (°C) | N[h] |
|---|---|---|---|---|---|---|---|---|---|---|---|---|
| StormVEx | 88 (116) | 2.4 (129) | 95 (132) | 91 | 3.7 | 0.090 (118) | 211 (54) | 0.117 (63) | 9.2 (22) | 6.1 (30) | -12.8 (22) | 2955 |
| IFRACS | 123 (146) | 5.9 (112) | 141 (142) | 83 | 7.2 | 0.193 (109) | 199 (73) | 0.126 (54) | 10.1 (27) | 6.0 (35) | -8.2 (44) | 2580 |

[a]CIP concentration from 75-200 μm

[b]CIP concentration ≥400 μm

[c]CIP concentration ≥75 μm

[d]Ice water content

[e]Cloud droplet number concentration

[f]Cloud liquid water content

[g]Cloud droplet number-weighted mean diameter

[h]Number of 1 minute observations in the average





**Table 2**. Frequency distribution of Conc75-200, wind speed, and temperature as a function of wind direction.

| | StormVEx | | | | IFRACS | | | |
|---|---|---|---|---|---|---|---|---|
| Wind Direction (degrees) | Conc75-200 (L$^{-1}$) | Wind Speed (m s$^{-1}$) | Temp.[a] (°C) | Frequency[b] (%) | Conc75-200 (L$^{-1}$) | Wind Speed (m s$^{-1}$) | Temp. (°C) | Frequency (%) |
| >0-30 | - | - | - | - | 27 | 6.3 | -10.6 | 0.194 |
| >180-210 | 18.2 | 5.7 | -9.3 | 1.2 | 28 | 5.0 | -9.6 | 4.7 |
| >210-240 | 73 | 5.2 | -10.7 | 9.0 | 56 | 7.5 | -9.9 | 22.4 |
| >240-270 | 79 | 9.1 | -11.7 | 14.5 | 72 | 8.9 | -6.9 | 15.6 |
| >270-300 | 92 | 5.7 | -13.5 | 58.8 | 146 | 8.3 | -8.1 | 29.1 |
| >300-330 | 66 | 3.4 | -13.2 | 15.1 | 190 | 6.7 | -7.5 | 23.0 |
| >330-360 | 460 | 7.8 | -10.7 | 1.4 | 231 | 7.6 | -7.9 | 5.0 |

[a]Temperature based on 5 minute measurements.

[b]Based on the number of minutes in Table 1.





**Table 3**. Relationships among TAS, Conc75-200 (small crystals), and CDNC25-35 (large droplets) during StormVEx and IFRACS. r is the Pearson (Spearman Rank) correlation.

| | StormVEx | | IFRACS | |
|---|---|---|---|---|
| TAS (m s$^{-1}$) | Conc75-200 (L$^{-1}$) | N | Conc75-200 (L$^{-1}$) | N |
| 1-3 | 39 | 51 | 46 | 111 |
| 3-5 | 51 | 928 | 49 | 800 |
| 5-8 | 84 | 1463 | 112 | 1258 |
| 8-12 | 175 | 513 | 301 | 382 |
| 12-16 | - | - | 616 | 29 |
| r | 0.38 (0.36) | | 0.54 (0.47) | |

| | StormVEx | | IFRACS | |
|---|---|---|---|---|
| CDNC25-35 (L$^{-1}$) | Conc75-200 (L$^{-1}$) | N | Conc75-200 (L$^{-1}$) | N |
| 0-25 | 70 | 1369 | 24 | 488 |
| 25-50 | 117 | 407 | 41 | 501 |
| 50-100 | 172 | 216 | 62 | 434 |
| 100-500 | 223 | 202 | 154 | 775 |
| 500-2000 | 228 | 17 | 379 | 339 |
| >2000 | - | - | 480 | 21 |
| r | 0.24 (0.33) | | 0.61 (0.68) | |



**Table 4**. Results of stepwise regression[a] of Conc75-200 on the explanatory variables TAS, CDNC25-35, and temperature (Temp). Variables are entered according to their contributions to the model r square. Partial r square are shown for each variable. Pr>F is the probability that the estimated contribution to the model is random. Importance[b] is a metric of the non parametric regression procedure which is analogous to the partial r square.

| | StormVEx | | IFRACS | |
| --- | --- | --- | --- | --- |
| | Partial r-square | Importance | Partial r-square | Importance |
| TAS | 0.188 | 100 | 0.131 | 44 |
| CDNC25-35 | 0.051 | 63 | 0.31 | 100 |
| Temp | 0.0014 | 5.7 | 0.035 | 21 |

[a]Stepwise regression was done using the REG procedure with forward selection in SAS 9.4.

[b]Non parametric regression was done using the ADAPTIVEREG procedure in SAS 9.4.



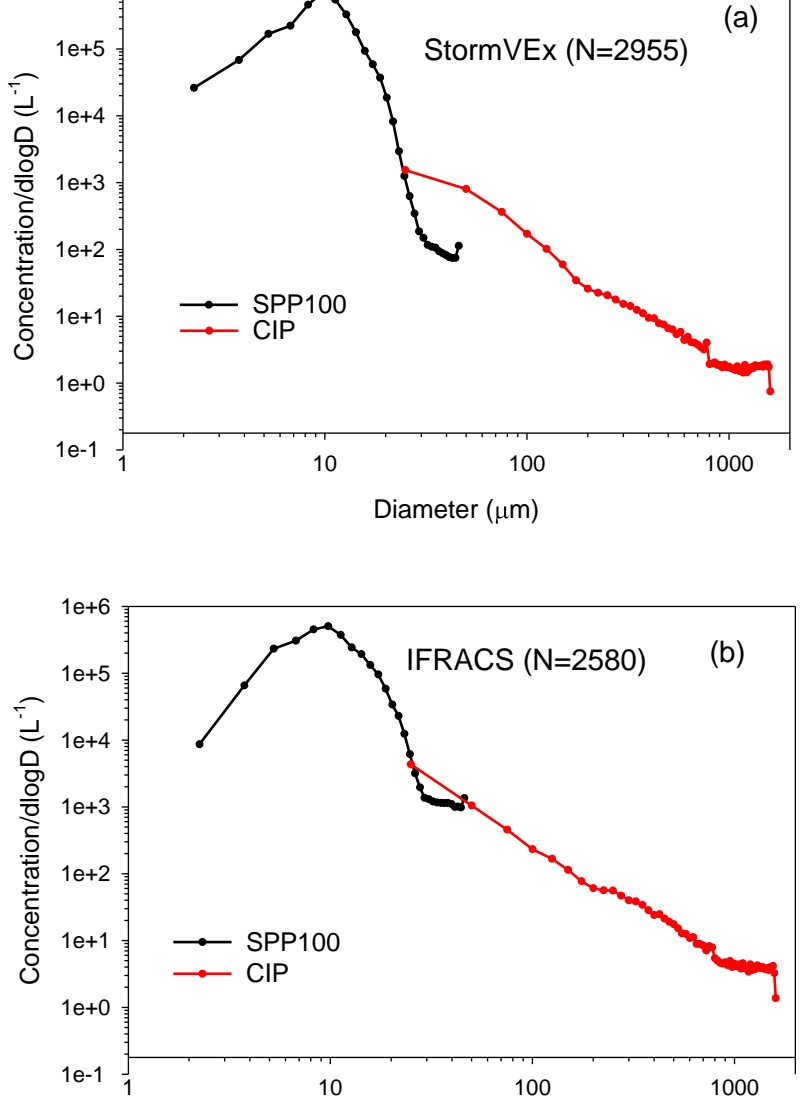

**Figure 1**. Average of concurrent 1 minute SPP-100 and CIP particle size distributions (PSDs) from StormVEx (a) and IFRACS (b).





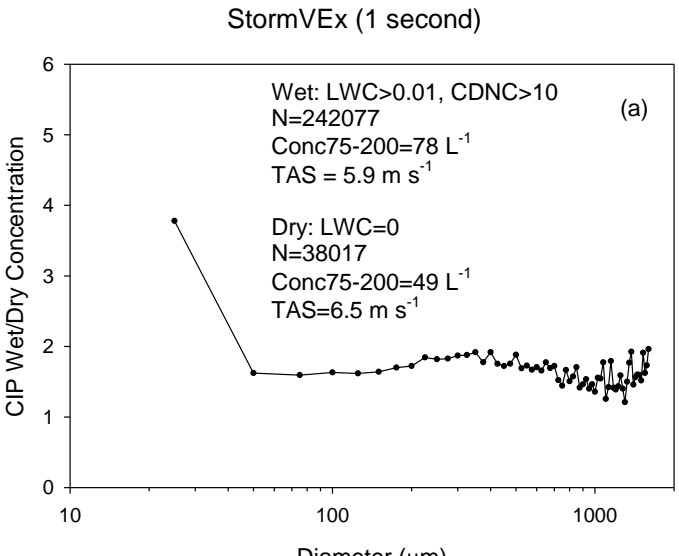

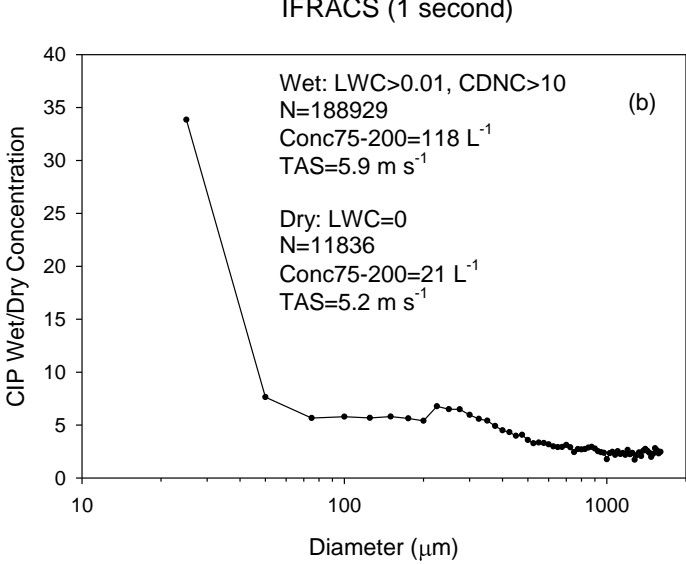

**Figure 2.** Ratio of average mixed phase (LWC>0.01 g m$^{-3}$, CDNC>10 cm$^{-3}$) to dry (LWC=0) PSDs for StormVEx (a) and IFRACS (b).




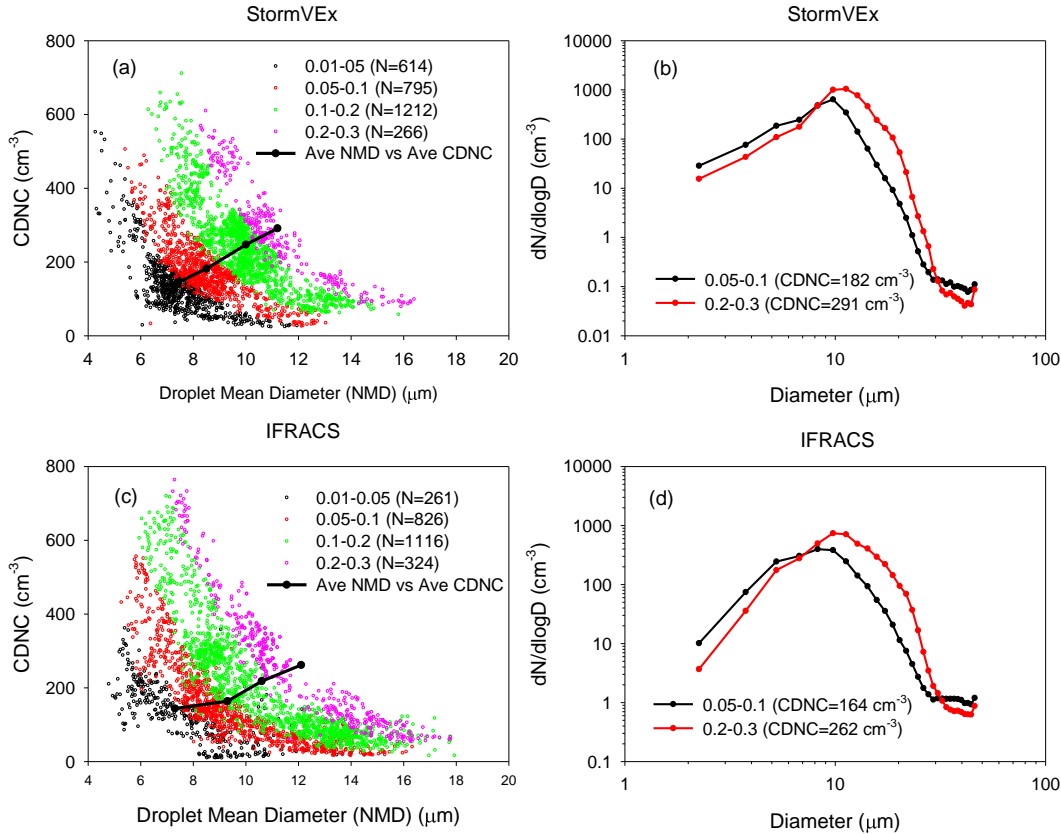

**Figure 3**. Relationships among 1 minute average mean cloud droplet diameter (NMD), concentration (CDNC), and liquid water content (LWC, µg m$^{-3}$) during StormVEx (a) and IFRACS (c). Corresponding average PSDs for low (0.01-0.05 g m$^{-3}$) and high (0.2-0.3 g m$^{-3}$) LWC are shown in Figs. 3b and 3d.



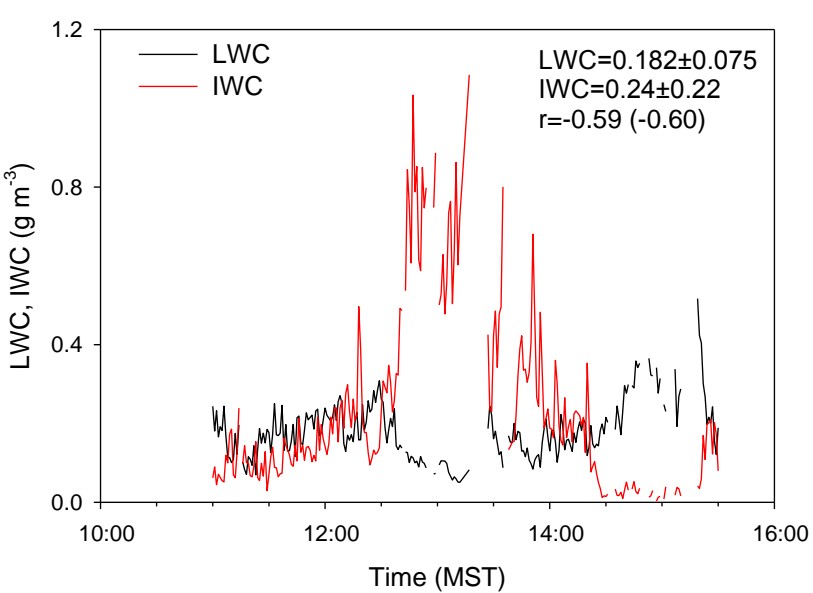

**Figure 4**. Time series of LWC and IWC on February 9, 2014 during IFRACS.





**Figure 5**. Average PSDs for high (1245-1317 MST) and low (<1245 or >1317 MST and LWC/IWC>2) IWC periods in Fig. 4. The values in the middle of the plot are the differences between the high (red) and low (black) cumulative LWC in the three sections of the distributions defined by the vertical dotted lines.




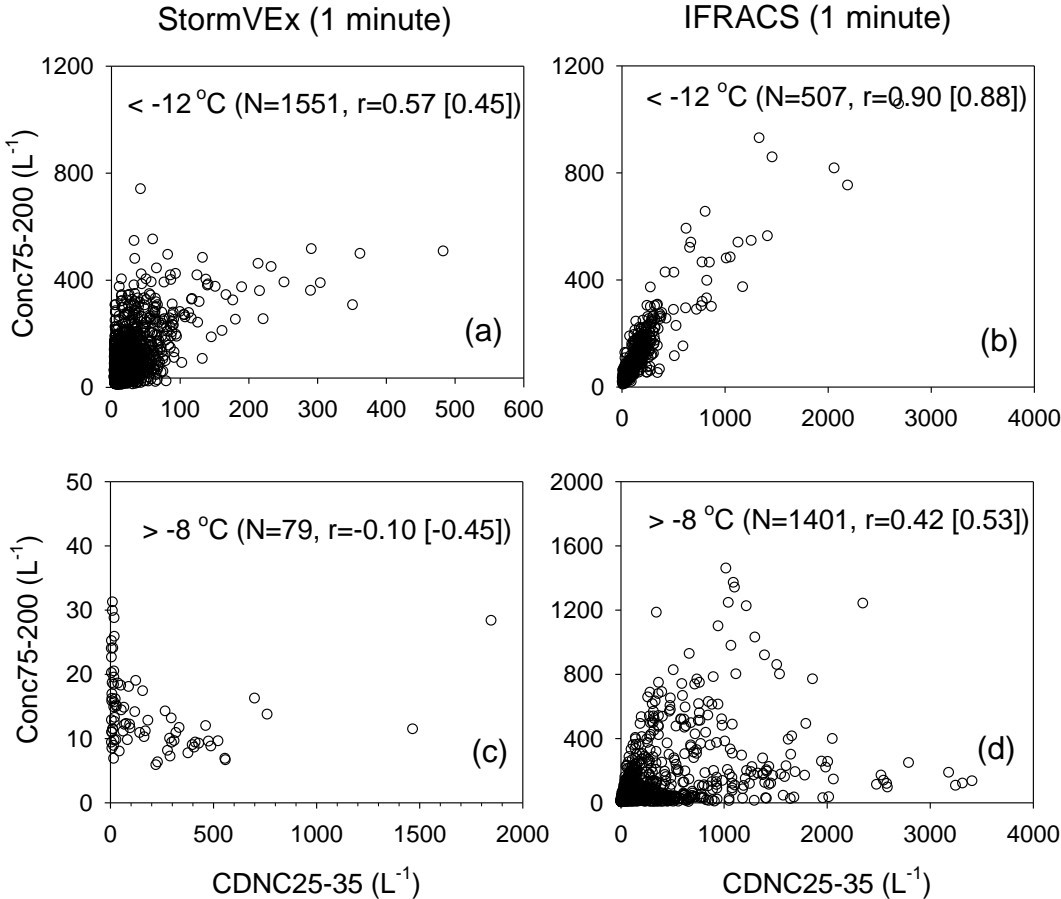

**Figure 6**. Relationships between 1 minute average concentrations of large cloud droplets (25-35 μm, CDNC25-35) and small crystals (Conc75-200) during (a) StormVEx and (b) IFRACS under cold conditions (<12 °C). Corresponding relationships under warm (>-8 °C) conditions are shown in (c) and (d). Number of observations and Pearson (Spearman Rank) correlations are shown.



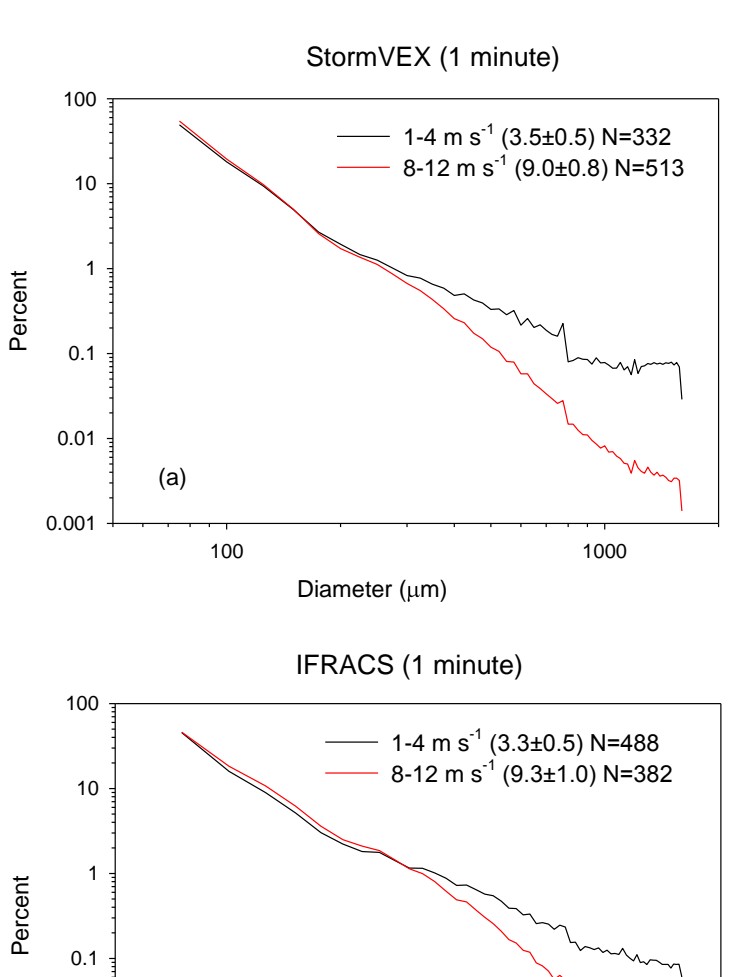

**Figure 7**. Averages of 1 minute relative (%) CIP PSDs at low (1-4 m s⁻¹) and high (8-12 m s⁻¹) TAS during StormVEx (a) and IFRACS (b). Average TAS are given in parentheses.