# Peer review of "Mixed-phase Orographic Cloud Microphysics during StormVEx and IFRACS"

_Atmospheric Chemistry and Physics, 2018_

## Referee Comment (RC1) · Anonymous Referee #1 · 19 Nov 2018

Review of "Mixed Phase Orographic Cloud Microphysics during Storm Vex and IFRACS"

The authors describe cloud microphysical measurements made at Storm Peak Laboratory (SPL) during 2 measurement campaigns during 2011 and 2014. They used in-situ microphysics probes to measure cloud particles including ice crystals and liquid droplets and looked at relationships between the different cloud properties but also with ambient meteorological conditions.

They find some already established results, for example an inverse relationship between cloud droplet size and concentration due to CCN availability, and some less well understood results such as a correlation between large drops and small ice crystals and enhancement in ice crystal concentrations in general.

[Figure]

The paper has the potential to provide useful results but I suggest major corrections are needed before the paper is considered for publication in ACP.

Major Comments

I found the descriptions of the experimental setup to be confusing, particularly the explanation of how the instruments were aspirated, orientated into the wind and the steps taken to quality control the measurements made by instruments subject to the harsh environmental conditions. Any revision should include a new figure detailing the different instruments and the setup at SPL and a more in depth discussion of how the probes were aspirated.

The most novel measurements relate to the enhanced ice crystal concentrations at a mountain top site like this where the cloud is in contact with the surface and interaction processes are poorly understood. The CIP-25 probe is key to this – there should be presentation of some of the imagery from the instrument and a detailed justification of why using a size threshold >62.5um for ice was appropriate. The data is sometimes presented in a confusing way. There are numerous times the data is compared for warm and cold conditions. Cold is defined as below -12 and warm above -8. What happens in between this range?

I didn't find the interpretation of some of the results very convincing. It is proposed that the relationship between large drops and small ice crystals could be due to immersion or contact freezing. There is a relationship found between larger supercooled drops and ice crystal concentrations but I'm not sure how the jump is made to the impact of these bigger drops being increased appearance of small ice crystals through primary ice nucleation when it would seem more likely to be a secondary process of some kind.

One of the key suspects (but not the only one) for enhanced ice crystal concentrations in supercooled orographic clouds in contact with a frozen surface is some process that provides ice from the surface. The mechanism by which this might happen is still very uncertain but I'd like to see a bit of information about the topography and surfaces

around SPL.

I think the relevant ideas and literature are generally included but the results are framed poorly. My interpretation is that the main findings surround the enhanced ice crystal concentrations vs what you might expect at these temperatures, but I felt that although the different potential mechanisms for the production of these were stated, the authors didn't present coherent conclusions.

Minor Comments:

1. In the abstract I suggest removing acronyms that don't appear in the abstract again and then defining them in the body of the manuscript.

2. StormVEx and IFRACS should be defined in the abstract

3. P2 L35 is second reference in brackets correct?

4. P3 L86 I may have missed it earlier but if not please define DRI.

5. P3 L93 variation over a 3 year period? If the campaign periods are correct it isn't over 3 years but you do compare between the two time periods (Nov 2010 – Apr 2011 and Jan – Feb 2014).

6. P3 L106 Add FSSP acronym

7. P3 107 the SPP-100 acronym might refer to the electronics revision of the FSSP. I'd prefer this instrument to be referred to as the Forward Scattering Spectrometer Probe (FSSP)

8. P4 L108 who is the manufacturer?

9. P4 L109 What is the face velocity?

10. P4 L142 If the TAS vs the OAP set airspeed is not equal you begin to get distortions of aspect ratio that will lead to changes in size. I think it is inaccurate to state that the misshapen particles will not necessarily be sized incorrectly.

11. P5 L164 I'd be interested to know with these constraints which contributes to the loss of ∼ 50 % of data. You say the CIP measures particles for 101.4 and 77.2 hours respectively, so condition 1 is met for this number of hours. You are then left with 49.2 and 43 hours of MPC suggesting you were in glaciated for considerable time periods (condition 2 and 3 not met but condition 1 is true).

12. Section 3.1 header has SPP-100 I might be getting this wrong but I think this should be the FSSP

13. P5 L178 the nominal size range quotes is below the threshold you stated earlier in the paper.

---

## Referee Comment (RC2) · Anonymous Referee #2 · 22 Nov 2018

The authors present the analysis of an observation of orographic mixed-phase clouds with ground-based in-situ instruments at the Storm Peak Laboratory. The 92 hours data were analysed in a statistical way to explore relationships between microphysical properties and draw conclusion about the ice crystal formation processes.

The impressive dataset of 92 hours of mixed-phase cloud measurement is relevant for a publication and it fits in the scope of ACP, but I suggest major correction are needed before publishing this paper. A large part of the argumentation uses the cloud particle concentration between 25-100 $\mu$m. This is a delicate size range. On the one hand, the CIP have a larger uncertainty in the smallest sizes bins because of diameter corrections. On the other hand, it cannot be assumed that the transition between liquid droplets and ice crystals is under all condition at the same size range. A more thorough

discussion of the uncertainties and assumptions would strengthen the argumentation of the paper.

For some argumentation, the interpretation of the data is inconclusive. Why is a 10-fold increase unrealistic, although the MTAS it doubled? Could the relationships between large cloud droplets (CDNC25-35) and small ice crystals (Conc75-200) also caused because a large percentage of the CDNC25-35 are ice crystals? Can a relative enchainment of small ice crystals be excluded using the relative PSDs in Figure 7?

Specific comments:

Line 119 – 121: A more detailed discussion about the setup, including a picture, would be beneficial to discuss the influence of the local surrounding including building. In particular, if the high Conc75-200 in the NNW sector (discuss in Line 210-214) could be due influence of the railing, terrace, etc.

Figure 3 right side and Figure 5: To decide if the difference is significant, I suggest an estimate of the measurement uncertainty (either by error bars or by a discussion of the measurements uncertainty).

Line 229 and Figure 5: Could the particles larger than 35 $\mu$m also be ice crystals?

Line 235-237: How can from Fig. 1 and 4 concluded that the liquid cloud was not effected by the ice phase?

Line 241 – 243: Without the plot, it is hard to follow and visualize the argumentation. Consider to include the plot in the paper?

Figure 5: Shading of the times used for the low/high IWC analysis would increase comprehensibility.

Line 259 – 263: I suggest giving the two cases a clearer name, e.g. Low-Ice and High-Ice. I was confused that in line 259 – 263 concentration where give behind IWC and thought for a while that with IWC is the acronym of ice water concentration.

Line 261-262: Could the particles larger than 35 $\mu$m also be ice crystals?

Line 269 – 273: As the amount of blowing snow non-linear increases with the wind speed, I would assume that the MTAS is more relevant for the amount blowing snow particles. I find a 10-fold increase not unrealistic, in particular as in Beck et al., 2018, a case is shown in the upper panel of figure 9, where a 10-fold increase is measured above a sharp threshold wind speed.

Line 275 – 279: Could also a higher percentage of the CDNC25-35 be ice crystals than indicated by the wet/dry ratio? During wet condition, processes like hoar/surface frost (Lloyd et al., 2015) could have produced more small ice crystals or a stronger overestimation of the concentration in smallest bin of the SSP due to wrongly correction of the size of ice particles. A look at the overlap of in the size distribution (similar to Fig. 1) stratified by wet and dry condition could help to understand it.

Line 333 – 340: In my view, there is a misunderstanding in some of the conclusion from the Beck et al., 2017 paper. In the Beck et al., 2017, paper the authors conclude that ICNCs decrease with height. ICNCs near the ground are at least a factor of 2 larger also when the SBO was in liquid clouds.

Line 347 – 354: If the CDNC25-35 were dominated by ice crystal produced by a ground-based process depending on the TAS, the high importance of CDNC25-35 could be also a consequence of the TAS.

Line 356 – 363: The authors want to find a relative enchainment of small ice crystals due to blowing snow by using the relative PSDs of low and high wind cases (Figure 7). What effect would a relative a relative enchainment of small ice crystals have on Figure 7? In my understanding, the relative PSD would slightly increase for small diameter (because an increase of small ice crystal concentration, but also an increase in the total concentration) and would lead to a stronger decrease for large diameter (because the big ice does not increase much, but the total concentration increase). This changes, where observed in Figure 7. Maybe the author could show how different

PSD would translate to differences in the relative PSD of Figure 7?

Technical Corrections/Minor Comments

Line 17: to be consistent: "cloud condensation nuclei"

Line 40: write out acronym "mixed-phase clouds (MPC)"

Table 1: Unclear where the parameters measured by the CIP stops and where the SPP-100 parameter starts. For consistency write out TAS.

Line 182: To be consistent with units it should be "sampling flow speed"

Line 183: In my understanding with higher flow speeds in the inlet than outside you have superisokinetic sampling, which leads to an undersampling of the large particles and not an oversampling of smaller droplets.

Line 198 – 207: This part was hard to follow and a rephrasing might help. If the first two CIP channels cannot be trusted, which I agree to, than the argumentation might be obsolete.

Figure 3: Mention that the colors on the left side are the liquid water contents

Line 240: The sentence "Henceforth, the Spearman Rank correlation is displayed in parenthesis after the Pearson correlation." is confusing, in particular as the Spearman Rank is not in parenthesis two sentence before.

Line 286: In the text is written "29 data points" are used, but in Figure 6 N=79.

Line 296: I think it should be "IFRACS (Figure 6b)."

---

## Author Comment (AC1) · 14 Feb 2019

The reviewer's comments are presented in italics, followed by our responses. We thank the reviewer for his comments.

*The authors describe cloud microphysical measurements made at Storm Peak Laboratory (SPL) during 2 measurement campaigns during 2011 and 2014. They used in-situ microphysics probes to measure cloud particles including ice crystals and liquid droplets and looked at relationships between the different cloud properties but also with ambient meteorological conditions. They find some already established results, for example an inverse relationship between cloud droplet size and concentration due to CCN availability,*

While the relationship between drop size and concentration is expected, the original Fig. 3 presents an added dimension to this relationship, which is found to depend on liquid water content. This demonstrates a unique aspect and a potential drawback to studies at mountaintop laboratories, where the cloud base height may vary below the lab.

*and some less well understood results such as a correlation between large drops and small ice crystals and enhancement in ice crystal concentrations in general. The paper has the potential to provide useful results but I suggest major corrections are needed before the paper is considered for publication in ACP.*

*Major Comments*

*I found the descriptions of the experimental setup to be confusing, particularly the explanation of how the instruments were aspirated, orientated into the wind and the steps taken to quality control the measurements made by instruments subject to the harsh environmental conditions. Any revision should include a new figure detailing the different instruments and the setup at SPL and a more in depth discussion of how the probes were aspirated.*

The probe setup was discussed in lines 104-121. A new figure (Fig. 1 (revised), below) was added showing the probe configuration and a view upwind. The last sentence in this section starting on line 119 was moved to the end of line 104: "The cloud probes were mounted on a rotating wind vane (to orient them into the wind) located on the west (upwind) railing of the roof approximately 6 m above the snow surface. The sentence "The cloud probes were calibrated and serviced at DMT prior to each field campaign." was moved to the end of the section (line 121). The revised sentence notes that calibration and servicing was done at DMT. The discussion of how the probes were "aspirated" pertains only to the FSSP-100, which is fitted with a fan that draws air through the probe's sample volume. The CIP is not aspirated. The CIP sample volume is based on the wind speed measured with a sonic anemometer. This was clearly described in the text.

*The most novel measurements relate to the enhanced ice crystal concentrations at a mountain top site like this where the cloud is in contact with the surface and interaction processes are poorly understood. The CIP-25 probe is key to this – there should be presentation of some of the imagery from the instrument and a detailed justification of why using a size threshold >62.5um for ice was appropriate.*

Figure 8 (revised), below, which is included in the revised manuscript, shows CIP images from the case study on 9 February 2014 (discussed in section 3.3). The following is inserted into the text on line 252: "Figures 8a and 8b present CIP images from the high and low IWC periods, respectively. Note the relatively higher concentration of "dots" in Fig. 8b (low IWC, high LWC). These represent cloud droplets that occluded a single CIP diode."

*and a detailed justification of why using a size threshold >62.5um for ice was appropriate.*

This is discussed in the paragraph beginning on line 191. The objective was to exclude cloud droplets from the CIP data. There is no evidence for droplets this large in these shallow, orographic clouds when SPL is close to cloud base. Indeed, Lloyd et al. (2015) concluded that there were no droplets larger than 35 µm in orographic clouds at the Jungfraujoch. The original Fig. 2, which compares the CIP size distributions for wet (LWC>0) and dry (LWC=0) cases during StormVEx and IFRACS, demonstrates that only the first CIP size bin was impacted by droplets.

*The data is sometimes presented in a confusing way. There are numerous times the data is compared for warm and cold conditions. Cold is defined as below -12 and warm above -8. What happens in between this range?*

The logic was that if a process is temperature-dependent, that process will be observed most readily at extremes in temperature. In this case, 93 and 81% of 1-minute average observations occurred at temperatures lower than -10 ℃ during StormVEx and IFRACS, respectively. The relationship between large drops and small crystals is similar at cold temperatures defined as either <-12 ℃ or <-8 ℃. In response to this question, "cold" is defined as <-8 ℃ in the revised manuscript.

*I didn't find the interpretation of some of the results very convincing. It is proposed that the relationship between large drops and small ice crystals could be due to immersion or contact freezing. There is a relationship found between larger supercooled drops and ice crystal concentrations but I'm not sure how the jump is made to the impact of these bigger drops being increased appearance of small ice crystals through primary ice nucleation when it would seem more likely to be a secondary process of some kind. One of the key suspects (but not the only one) for enhanced ice crystal concentrations in supercooled orographic clouds in contact with a frozen surface is some process that provides ice from the surface. The mechanism by which this might happen is still very uncertain but I'd like to see a bit of information about the topography and surfaces around SPL. I think the relevant ideas and literature are generally included but the results are framed poorly. My interpretation is that the main findings surround the enhanced ice crystal concentrations vs what you might expect at these temperatures, but I felt that*

*although the different potential mechanisms for the production of these were stated, the authors didn't present coherent conclusions.*

The hypothesis that ice crystal production is related to large drops was raised in previous studies, as described in lines 70-74 of the original manuscript. Our analysis shows a relatively strong relationship between large droplet and small ice crystal concentration at low temperatures during IFRACS. This is **new evidence** for heterogeneous freezing of large droplets, whether by immersion or contact freezing. We see nothing logically or scientifically incoherent about this conclusion. On the other hand, the reviewer states that "it would seem more likely to be a secondary [ice production] process [SIP] of some kind". Upon what evidence is this belief based? There is a lot of speculation about SIP in the literature but in the absence of direct evidence from our studies, we prefer not to engage in it.

*Minor Comments:*

*1. In the abstract I suggest removing acronyms that don't appear in the abstract again and then defining them in the body of the manuscript. 2. StormVEx and IFRACS should be defined in the abstract.*

In the revised manuscript, all acronyms are defined at their first use in both the abstract and main text.

*3. P2 L35 is second reference in brackets correct?*

Corrected to Peng.

*4. P3 L86 I may have missed it earlier but if not please define DRI.*

Desert Research Institute inserted on L85.

*5. P3 L93 variation over a 3 year period? If the campaign periods are correct it isn't over 3 years but you do compare between the two time periods (Nov 2010 – Apr 2011 and Jan – Feb 2014).*

Changed "over a 3-year period" to "over two winters".

*6. P3 L106 Add FSSP acronym*

*7. P3 107 the SPP-100 acronym might refer to the electronics revision of the FSSP. I'd prefer this instrument to be referred to as the Forward Scattering Spectrometer Probe (FSSP)*

*8. P4 L108 who is the manufacturer?*

In all cases, "SPP-100" is replaced by FSSP-100, noting that the electronics were modified by DMT. The aspirator was purchased with the original PMS FSSP-100 probe, as noted in the revised manuscript.

*9. P4 L109 What is the face velocity?*

It's the velocity at the center of the inlet.

*10. P4 L142 If the TAS vs the OAP set airspeed is not equal you begin to get distortions of aspect ratio that will lead to changes in size. I think it is inaccurate to state that the misshapen particles will not necessarily be sized incorrectly.*

There is no "set" air speed for the CIP. The air speed (TAS) is supplied by the sonic anemometer, as described in lines 116-117. As noted in the manuscript, we used the "area equivalent diameter" to size CIP particles. This is the diameter of a sphere with the same projected area of a particle which may be irregular. We intentionally used this method because distortions may arise from incorrect wind speed. The reviewer is correct that aspect ratios would be sensitive to such distortions but our analysis does not include this parameter.

*11. P5 L164 I'd be interested to know with these constraints which contributes to the loss of _ 50 % of data. You say the CIP measures particles for 101.4 and 77.2 hours respectively, so condition 1 is met for this number of hours. You are then left with 49.2 and 43 hours of MPC suggesting you were in glaciated for considerable time periods (condition 2 and 3 not met but condition 1 is true).*

The data completeness requirement (75%, line 163) accounts for the reduced number of 1-minute averages. LWC was zero during 10 and 4.3% of seconds when Ni was greater than zero during StormVEx and IFRACS, respectively. This is apparent from the numbers given in the original Fig. 2 and the total number of observations given in the text.

*12. Section 3.1 header has SPP-100 I might be getting this wrong but I think this should be the FSSP.*

Changed to FSSP, as noted above.

*13. P5 L178 the nominal size range quotes is below the threshold you stated earlier in the paper.*
*Interactive comment on Atmos. Chem. Phys. Discuss., [https://doi.org/10.5194/acp-2018](https://doi.org/10.5194/acp-2018)*

Optical array probes have been widely used for decades and a detailed explanation of how they work can found in the literature. 12.5-37.5 µm indicates the lower and upper bounds of the first CIP size channel given the 50% reduction of laser energy needed to trigger a diode. The size bins are typically referred to by their midpoints, i.e., the first channel is 25 µm and the 64th channel is 64x25=1600 µm. In presentations of size distributions, the bin boundaries are taken as the midpoint ±12.5 µm. The following sentence was added at line 115: "An array diode is triggered when a particle obscures >50% of the incident laser energy on the diode.".

**References**

Lloyd, G., Choularton, T.W., Bower, K.N., Gallagher, M.W., Connolly, P.J., Flynn, M., Farrignton, R., Crosier, J., Schlenczek, O., Fugla, J., and Henneberger, J.: The origins of ice crystals measured in mixed-phase clouds at the high-alpine site Jungfraujoch. Atmos. Chem. Phys., 15, 12953-12969, doi:10.5194/acp-15-12953-2015, 2015.

[Figure]

**Figure 1 (revised)**. Recent picture of SPL probe stand used previously during StormVEx and IFRACS with FSSP-100 and sonic anemometer on top and DMT CIP and PIP (Precipitation Imaging Probe) on left and right sides, respectively. View facing west over the railing (right panel).

[Figure]

(a)  (b)

**Figure 8 (revised)**. CIP images from 9 February 2014: (a) 13:12:19 MST, high IWC and low LWC, and (b) 12:29:09 MST, low IWC and high LWC. The vertical bars contain all of the images sampled in 1 second. The width of each bar corresponds to 1600 µm.

---

## Author Comment (AC2) · 14 Feb 2019

The reviewer's comments are presented in italics, followed by our responses. The authors thank the reviewer for his insight.

*The authors present the analysis of an observation of orographic mixed-phase clouds with ground-based in-situ instruments at the Storm Peak Laboratory. The 92 hours data were analysed in a statistical way to explore relationships between microphysical properties and draw conclusion about the ice crystal formation processes. The impressive dataset of 92 hours of mixed-phase cloud measurement is relevant for a publication and it fits in the scope of ACP, but I suggest major correction are needed before publishing this paper. A large part of the argumentation uses the cloud particle concentration between 25-100 um. This is a delicate size range. On the one hand, the CIP have a larger uncertainty in the smallest sizes bins because of diameter corrections. On the other hand, it cannot be assumed that the transition between liquid droplets and ice crystals is under all condition at the same size range. A more thorough discussion of the uncertainties and assumptions would strengthen the argumentation of the paper.*

One of the objectives was to test the hypothesis that ice particles were formed by heterogeneous freezing of large cloud droplets. Correlations between large droplet (25-35 µm, CDNC25-35) and small ice particle (75-200 µm, Conc75-200) concentrations were examined. In response to the reviewer's concerns, a closer look at the 1-second data revealed that the FSSP reported concentrations of 0 for particles larger than 25 µm for 86 and 44% of 1-second measurements during StormVEX and IFRACS, respectively. The paucity of non-zero CDNC25-35 during StormVEx is probably the result of a measurement issue related to the lack of the FSSP-100 scarf tube during that study. StormVEx is thus excluded from this analysis in the revised manuscript.

The reviewer rightly points out that it is not possible to observe the degree to which large droplets measured with the FSSP were actually small ice crystals. The implication is that if large droplets (25-35 µm) were actually ice crystals, the relationship between CDNC25-35 and Conc75-200 could simply represent autocorrelation between two segments of the ice crystal distribution. In the original Fig. 2, we demonstrated that on average, the CIP distribution was dominated by droplets in its smallest size bin, which corresponds to the large droplet size range. The reviewer suggests that while this may true in general, there may be periods when there is more ice in this size range than on average but it is also true that there may be periods when there is less ice than on average. Since our analysis is statistical in nature, variation in the proportions of liquid and ice in the 25-35 µm size range of the FSSP should manifest as noise in the relationship between CDNC25-35 and Conc75-200. Further, if large droplets were actually crystals, and this accounted for the relationship between CDNC25-35 and Conc75-200 at <-8 °C, why do we see not see this relationship at >-8 °C? In the revised manuscript, we acknowledge and discuss the limitations of the instrumentation in this regard and note that higher resolution instruments are needed to directly address these issues.

*For some argumentation, the interpretation of the data is inconclusive. Why is a 10-fold increase unrealistic, although the MTAS it doubled? Could the relationships between large cloud droplets (CDNC25-35) and small ice crystals (Conc75-200) also caused because a large percentage of the CDNC25-35 are ice crystals? Can a relative enchainment of small ice crystals be excluded using the relative PSDs in Figure 7?*

Correlation of wind speed with crystal concentrations during the day on 9 February 2014 does not necessarily imply blowing snow. The revised analysis of this case (see below) shows an inverse correlation between MTAS and Ni during the High- and Low-Ice periods. This behavior is not consistent with blowing snow. Production of ice is dependent on mesoscale and orographic dynamics (Neiman et al., 2002; Stoelinga et al., 2013). Higher wind speeds across the barrier produce more lifting and produce new ice particles upwind and above the barrier. Clearly, not all of the variation in ice crystal concentration at mountaintop laboratories is due to blowing snow although blowing snow does occur. We provide evidence against the blowing snow hypothesis but acknowledge in the revised Conclusions that it can be a factor: "It is possible that both primary production and blowing snow were active at SPL.".

A key point is that higher-resolution instruments such as holographic imagers used by Beals et al. (2015) (HOLODEC) and Beck et al. (2018) (HOLIMO) are needed to distinguish liquid droplets from ice crystals in mixed-phase clouds. This is noted in the text and at the end of the revised Conclusions.

*Specific comments:*

*Line 119 – 121: A more detailed discussion about the setup, including a picture, would be beneficial to discuss the influence of the local surrounding including building. In particular, if the high Conc75-200 in the NNW sector (discuss in Line 210-214) could be due influence of the railing, terrace, etc.*

A new Fig. 1 showing the probes on the wind vane and a view upwind of SPL is included in the revised text. There is no upwind railing, terrace, or building in any direction to the west. The wind data were re-examined to explain high Conc75-200 in the NNW sector. During StormVEx, mostly all of the NNW cases were on 22 January 2011. The 5-minute average wind direction was exactly the same (351.9º) for 3.5 hours. It is not likely that a 5-minute average value could be the same to a tenth of a degree for two consecutive 5-minute periods, much less eighteen. During IFRACS, many of the NNW wind directions exhibited the same value for thirty minutes or more. The reason is that the wind vane becomes iced by riming and doesn't move. The data were screened for repeated 5-minute wind speeds and these were eliminated. This reduced the number of 1-minute observations by 2 and 4.7% during StormVEx and IFRACS, respectively. Table 2 was modified accordingly. The largest difference was in the NNW sector (330-360º) during StormVEx, where Conc75-200 decreased from 460 to 150 $L^{-1}$.

*Figure 3 right side and Figure 5: To decide if the difference is significant, I suggest an estimate of the measurement uncertainty (either by error bars or by a discussion of the measurements uncertainty).*

The reviewer is referring to comparisons of FSSP PSDs in the original Figs. 3 and 5. The implication is that the differences in the PSDs at high and low LWC, which occurred in both 2011 and 2014, could have arisen by chance. This seems highly implausible. A statistical comparison of mean PSDs could be used to test the hypothesis that these apparent differences were not significant, i.e., they were fortuitous. In any case, for such a comparison, a simple t-statistic would be the difference divided by the standard error (standard deviation divided by the square root of the number of observations). The standard errors are plotted in the revised Figs. 4 and 7. They are very small because of the large number of measurements used to calculate the mean concentrations as a function of size.

*Line 229 and Figure 5: Could the particles larger than 35μm also be ice crystals?*

In general, the answer is yes, but as noted in below and in the revised text, measurements of large particles by the FSSP is problematic.

*Line 235-237: How can from Fig. 1 and 4 concluded that the liquid cloud was not effected by the ice phase?*

We do not conclude here that the FSSP distribution was *not* impacted by the ice phase. We stated that such interactions are not evident in the original Fig. 1. Figure 4 hadn't been introduced at this point and should not have been included in this sentence.

*Line 241 – 243: Without the plot, it is hard to follow and visualize the argumentation. Consider to include the plot in the paper?*

A new Figure 5 showing these scatter plots is included in the revised text.

*Figure 5: Shading of the times used for the low/high IWC analysis would increase comprehensibility.*

The high IWC period is between the black lines in the revised plot. The low IWC periods are shaded in grey in the revised plot.

*Line 259 – 263: I suggest giving the two cases a clearer name, e.g. Low-Ice and High-Ice. I was confused that in line 259 – 263 concentration where give behind IWC and thought for a while that with IWC is the acronym of ice water concentration.*

This section was revised, referring to "High-Ice" and "Low-Ice" periods. Characterization of Ni during the High- and Low-Ice periods has been clarified.

*Line 261-262: Could the particles larger than 35 µm also be ice crystals?*

Again, yes.

*Line 269 – 273: As the amount of blowing snow non-linear increases with the wind speed, I would assume that the MTAS is more relevant for the amount blowing snow particles. I find a 10-fold increase not unrealistic, in particular as in Beck et al., 2018, a case is shown in the upper panel of figure 9, where a 10-fold increase is measured above a sharp threshold wind speed.*

As noted in the Methods section, the probes at SPL are mounted in front of a railing 6 m above the snow surface. Winds did not pass over the deck or from the direction of the building during sampling periods. The reviewer suggests that high ice particle concentrations (Ni) should have accompanied or followed the maximum wind gusts (MTAS). Figure 10 (revised), below, plots 1-second MTAS (Fig. 10a) and the corresponding 1-minute average TAS (Fig. 10b) against 1-minute average Ni for High-Ice, Low-Ice, and all other (Intermediate-Ice) periods on 9 February 2014. MTAS was highly correlated with TAS [0.90 (0.90)] over the course of the day. The highest Ni correspond to the highest MTAS (and TAS) and *visa versa*. Average MTAS was 16.6±2.4, 8.9±2.0, and 11.3±2.8 m s$^{-1}$ during High-, Low-, and Intermediate-Ice periods, respectively. This could imply that high Ni resulted from blowing snow when the winds were stronger in the early afternoon. Contrary to results reported in Fig. 9a in Beck et al. (2018), there was no clear MTAS threshold that produced a step function in Ni. In Fig. 10 (revised), there appears to be an inverse relationship between 1-minute MTAS and 1-minute Ni, especially for the High- and Low-Ice regimes. Beck et al. (2018) discussed how a relationship between MTAS and blowing snow could be averaged out if the averaging time is too long or obscured because of an [indeterminate] lag between the arrival of the gust and the particles that were lofted by it. Beck et al. (2018) suggested using an averaging time of 10-15 seconds. Figure 11 (revised), below, plots 15-second average (using the 75% data completeness criterion) Ni against MTAS for the High-, Low-, and Intermediate-Ice periods. The figure shows that while both MTAS and Ni varied considerably, there was no obvious wind speed threshold and the correlations between MTAS and Ni were actually negative under High- and Low-Ice conditions. These results are not consistent with the blowing snow hypothesis. This discussion and the new figures are included in the revised manuscript.

*Line 275 – 279: Could also a higher percentage of the CDNC25-35 be ice crystals than indicated by the wet/dry ratio?*

This is discussed in detail in response to the reviewer's introductory remarks. There could be more but also less ice particles in CDNC25-35 than indicated by the average wet/dry ratio, which suggests that CDNC25-35 represents

droplets. The fact that we see a relationship between CDNC25-35 and Conc75-200 at cold but not warm temperatures also suggests that CDNC25-35 represents droplets and not ice crystals.

*During wet condition, processes like hoar/surface frost (Lloyd et al., 2015) could have produced more small ice crystals or a stronger overestimation of the concentration in smallest bin of the SSP due to wrongly correction of the size of ice particles. A look at the overlap of in the size distribution (similar to Fig. 1) stratified by wet and dry condition could help to understand it.*

The suggestion that splintering of hoar frost on the snow surface leads to high Ni is speculation. There is no direct evidence in our data to support this. It is difficult to see how hoar frost on the surface would have been a source of ice particles since snow was always accumulating on the surface during the measurements at SPL. The most plausible surface-based mechanism for ice production is blowing snow, which is discussed at length.

*Line 333 – 340: In my view, there is a misunderstanding in some of the conclusion from the Beck et al., 2017 paper. In the Beck et al., 2017, paper the authors conclude that ICNCs decrease with height. ICNCs near the ground are at least a factor of 2 larger also when the SBO was in liquid clouds.*

This paragraph was replaced by the following sentence at line 331: "Beck et al. (2018) reported a large increase in Ni when the maximum wind speed increased from 14-16 to ≥16 m s$^{-1}$ at the Sonnblick Observatory in Rauris, Austria when winds were from the south.".

*Line 347 – 354: If the CDNC25-35 were dominated by ice crystal produced by a ground-based process depending on the TAS, the high importance of CDNC25-35 could be also a consequence of the TAS.*

See above. The reviewer keeps coming back to a hypothetical while the evidences suggests that in general, CDNC25-35 was not dominated by ice crystals.

*Line 356 – 363: The authors want to find a relative enchainment of small ice crystals due to blowing snow by using the relative PSDs of low and high wind cases (Figure 7). What effect would a relative a relative enchainment of small ice crystals have on Figure 7? In my understanding, the relative PSD would slightly increase for small diameter (because an increase of small ice crystal concentration, but also an increase in the total concentration) and would lead to a stronger decrease for large diameter (because the big ice does not increase much, but the total concentration increase).*

We assume that by "enchainment" the reviewer means "enrichment". First, Fig. 7 (revised Fig. 12) was modified to represent the same low wind speed range as presented in Table 3, i.e., 1-3 m s$^{-1}$ rather than 1-4 m s$^{-1}$. The result is that there was an enrichment in small crystals (Conc75-200) at high wind speed of 10% during StormVEx and 8%

during IFRACS. However, this difference is small compared to the factor of 4.5-6.5 increase in the absolute concentration of Conc75-200 at the higher wind speed. Ice crystal concentration is not simply a function of blowing snow but varies with synoptic and orographic dynamics, i.e., stronger uplift nucleates more crystals upwind and above the mountain barrier (e.g., Neiman et al., 2002; Stoelinga et al., 2013). In fact, there were moderately strong correlations between 1-minute horizontal and vertical wind speeds during StormVEx [0.75(0.72)] and IFRACS [0.66(0.67)].

*This changes, where observed in Figure 7. Maybe the author could show how different PSD would translate to differences in the relative PSD of Figure 7?*

It is unclear what the reviewer is asking for. The contrast in the original Fig. 7 was used to test a specific hypothesis, i.e., that blowing snow would be relatively enriched in small particles. Is the reviewer suggesting that we address a different hypothesis using relative PSDs?

*Technical Corrections/Minor Comments*

*Line 17: to be consistent: "cloud condensation nuclei"*

Done.

*Line 40: write out acronym "mixed-phase clouds (MPC)"*

Done.

*Table 1: Unclear where the parameters measured by the CIP stops and where the SPP-100 parameter starts. For consistency write out TAS.*

In the revised table, a vertical line separates CIP and FSSP parameters. TAS is defined as horizontal wind speed in the footnotes.

*Line 182: To be consistent with units it should be "sampling flow speed"*

Done, however, since the scarf tube accelerated the flow during IFRACS, the wording was changed to "sampling flow speed at the inlet".

*Line 183: In my understanding with higher flow speeds in the inlet than outside you have superisokinetic sampling, which leads to an undersampling of the large particles and not an oversampling of smaller droplets.*

The reviewer is correct. This could explain a loss of large droplets in the aspirated FSSP during StormVEx. Note that the flow accelerator (scarf tube) was used during IFRACS but not StormVEx. Gerber et al. (1999) discussed inertial concentration of large droplets in an aspirated FSSP fitted with a scarf tube (flow accelerator). Because the FSSP and CIP PSDs overlap more closely during IFRACS than StormVEx, the loss of large droplets due to super-isokinetic sampling may be somewhat offset by the inertial concentration of large droplets caused by the scarf tube. This is discussed in the revised text.

*Line 198 – 207: This part was hard to follow and a rephrasing might help. If the first two CIP channels cannot be trusted, which I agree to, than the argumentation might be obsolete.*

We agree and have removed this section.

*Figure 3: Mention that the colors on the left side are the liquid water contents*

Done in the revised figure heading.

*Line 240: The sentence "Henceforth, the Spearman Rank correlation is displayed in parenthesis after the Pearson correlation." is confusing, in particular as the Spearman Rank is not in parenthesis two sentence before.*

Line 240 begins with "Henceforth", meaning after line 240.

*Line 286: In the text is written "29 data points" are used, but in Figure 6 N=79.*
*Line 296: I think it should be "IFRACS (Figure 6b)."*

This section has been extensively revised, as discussed above.

**References**

Beals, M.J., Fugal, J.P., Shaw, R.A., Lu, J., Spuler, S.M., and Stith, J.L., Holographic measurements of inhomogeneous cloud mixing at the centimeter scale. Science, 350, 6256, 87-89, doi:10.1126/science.aab0751, 2015.

Beck, A., Henneberger, J., Fugal, J.P. David, R.O., Lacher, L., Lohmann, U.: Impact of surface and near-surface processes on ice crystal concentrations measured at mountain-top research stations. Atmos. Chem. Phys., 18, 8909-8927, https://doi.org/10.5194/acp-18-8909-2018, 2018.

Gerber, H., Frick, G., and Rodi, A.R.: Ground-based FSSP and PVM measurements of liquid water content. J. Atmos. Ocean. Technol., 16, 1143-1149, https://doi.org/10.1175/1520-0426(1999)016<1143:GBFAPM>2.0.CO;2.

Neiman P.J., Ralph, F.M., White, A.B., Kingsmill, D.E., and Persson, P.O.G., The statistical relationship between upslope flow and rainfall in California's coastal mountains: Observations during CALJET. Mon. Wea. Rev., 130, 1468–1492, 2002, https://doi.org/10.1175/1520-0493(2002)130<1468:TSRBUF>2.0.CO;2.

Stoelinga M.T., Stewart R.E., Thompson G., Thériault J.M., Microphysical processes within winter orographic cloud and precipitation systems. In: Chow F., De Wekker S., Snyder B. (eds) Mountain Weather Research and Forecasting. Springer Atmospheric Sciences. Springer, Dordrecht, https://doi.org/10.1007/978-94-007-4098-3_7, 2013.

9 February 2014 (1-minute)

[Figure]

[Figure]

**Figure 10 (revised)**. Relationships between 1-minute average TAS (a) and maximum 1-second TAS (MTAS) (b) and Ni for High-Ice, Low-Ice, and Intermediate-Ice (all other 1-minute periods) periods on 9 February 2014.

[Figure]

**Figure 11 (revised)**. Relationships between 15-second average Ni and MTAS for High-Ice (a), Low-Ice (b), and Intermediate-Ice (c) periods on 9 February 2014.